

# Efficient security level in wireless sensor networks (WSNs) using four-factors authentication over the Internet of Things (IoT)

Albandari Alsumayt[1], Majid Alshammari[2], Zeyad M. Alfawaer[1], Fahd N. Al-Wesabi[3], Nahla El-Haggar[1], Sumayh S. Aljameel[4], Sarah Albassam[5], Shahad AlGhareeb[5], Nouf Mohammed Alghamdi[5] and Nawir Aldossary[5]

[1] Department of Computer Science, Applied College, Imam Abdulrahman Bin Faisal University, Dammam, Saudi Arabia
[2] Department of Information Technology, College of Computers and Information Technology, Taif University, Taif, Saudi Arabia
[3] Department of Computer Science, College of Science & Art, King Khalid University, Mahayil, Saudi Arabia
[4] Saudi Aramco Cybersecurity Chair, Department of Computer Science, College of Computer Science and Information Technology, Imam Abdulrahman Bin Faisal University, Dammam, Saudi Arabia
[5] Saudi Aramco Cybersecurity Chair, Department of Networks and Communications, College of Computer Science and Information Technology, Imam Abdulrahman Bin Faisal University, Dammam, Saudi Arabia

Corresponding author
Albandari Alsumayt,
afaalsumayt@iau.edu.sa

## ABSTRACT

With the increasing demand for the use of technology in all matters of daily life and business, the demand has increased dramatically to transform business electronically especially regards COVID-19. The Internet of Things (IoT) has greatly helped in accomplishing tasks. For example, at a high temperature, it would be possible to switch on the air conditioner using a personal mobile device while the person is in the car. The Internet of Things (IoT) eases lots of tasks. A wireless sensor network is an example of IoT. Wireless sensor network (WSN) is an infrastructure less self-configured that can monitor environmental conditions such as vibration, temperature, wind speed, sound, pressure, and vital signs. Thus, WSNs can occur in many fields. Smart homes give a good example of that. The security concern is important, and it is an essential requirement to ensure secure data. Different attacks and privacy concerns can affect the data. Authentication is the first defence line against threats and attacks. This study proposed a new protocol based on using four factors of authentication to improve the security level in WSN to secure communications. The simulation results prove the strength of the proposed method which reflects the importance of the usage of such protocol in authentication areas.

## INTRODUCTION

In wireless sensor networks (WSNs), no fixed infrastructures or centralized access points are required to communicate over the shared wireless channels. It is one of the most promising

wireless technologies that employ the use of multi-hops to transmit information, creating one of the most promising sensor networks. There are many applications for WSNs, including in agriculture, industrial automation, transportation, military surveillance, environmental monitoring, healthcare, process monitoring, and many other areas. To communicate sensor information to a centralized control location, these wireless sensors must be self-configured into a network (*Rashid & Rehmani, 2016*). A traditional WSN typically consists of routers and switches, Consequently, monitoring and updating them becomes more difficult as they grow. The use of different communication protocols in large WSNs also contributes to their heterogeneity, which means that they are essentially composite networks that communicate on a very low level (*Kobo, Abu-Mahfouz & Hancke, 2017*). It is very complex to determine how security mechanisms will be applied in the network when a communication protocol is distributed, and nodes are assigned to receive or transmit data depending on the management of the protocol. In addition, as WSNs grow in scale, they are subject to several constraints, such as energy and resource limitations, processing and memory constraints, and communication limitations. To overcome these constraints, lightweight security frameworks for which intelligent features can be centrally centralized are necessary.

As one of the fundamental requirements of WSN applications, security must be implemented across a variety of types. Due to the hostile environments in which WSNs are deployed, they are particularly vulnerable to a variety of security attacks (*Tomić & McCann, 2017*). As all nodes transfer their data to the base station, WSNs are vulnerable to extra vulnerabilities (*Salmi & Oughdir, 2023*). As a result, WSNs are vulnerable to two types of attacks: outsiders and insiders. External attackers aim to corrupt the functionality of the network through injections into the network by external entities (*Silva, Zabaleta & Arizmendi, 2022*). Attacks by insiders involve the penetration of sensor nodes and the use of those nodes to launch attacks against another domain or to activate another attack (*Alansari et al., 2022*). An active routing disruption attack on the network layer is a sinkhole or blackhole attack, which is a major insider attack (*Shanmugaraja et al., 2023*). Attackers advertise themselves as high-quality paths to base stations where the nodes are closer to the base stations than others to attract other nodes (*Zhang et al., 2023a*). Consequently, malicious nodes utilize the malicious node path more frequently, thereby altering, spoofing, or dropping transmitted packets, and preventing base stations from receiving accurate or complete data (*Rehman, Rehman & Raheem, 2019*). As well as enabling other types of attacks, such as wormhole attacks and selective forwarding attacks, sinkhole attacks can be highly detrimental to WSNs (*Terence & Purushothaman, 2019*). However, despite their successful use in data integrity and authentication in many types of networks (public-key and private-key cryptography), traditional security mechanisms cannot be adopted in WSNs since they require more computational power and consume more energy, thereby reducing network lifespan (*Oladipupo et al., 2023*). WSNs should be secured by techniques that do not compromise the network's lifespan, thus preventing security from compromising the network's life.

It is fundamentally challenging to establish an authenticated key exchange *via* a gateway between a sensor and a user to prevent unauthorized access to sensitive sensor data. In

WSNs, authenticated key exchange is generally regarded as a more challenging task due to sensor resource constraints and network characteristics such as unattended operation and an unreliable communication channel, which makes it more challenging than achieving it in traditional networks with sufficient computing resources and existing infrastructure. When anonymity is desired, authenticated key exchange becomes even more challenging. In recent years, increasing awareness of privacy has led to users' anonymity becoming a vital security feature in a variety of WSN applications as well as many other applications, including location-based services, e-voting, mobile roaming, and anonymous web browsing.

The two-factor authentication scheme proposed by Das (*Chen et al., 2023*) uses a smart card and password to verify the identity of the user. Following this, many schemes are presented to improve it (*Zhao et al., 2020*; *Wazid et al., 2019*; *Bilal & Kang, 2017*; *Wang, Xu & Sun, 2017*; *Xu & Wu, 2019*; *Yu & Park, 2022*). Under the assumption that an adversary can extract parameters from a smart card that's lost or stolen, *Chen & Chen (2021)* identified several remaining security weaknesses. Following that, a new scheme of mutual authentication and key agreement was presented. For hierarchical wireless sensor networks, *Das et al. (2016)* and *Sahoo et al. (2023)* propose a two-factor authentication scheme. As a result of identifying the shortcomings of the previous authentication schemes, Yoo et al. (*Singh et al., 2020*) proposed a robust user authentication scheme. However, they fail to consider user privacy protections in their scheme. A study by Sun et al. (*Yang et al., 2020*) found that Khan and Alghathar's scheme is vulnerable to three types of attacks: GWN impersonation, GWN bypassing, and privileged insider attacks. Their proposal also includes a new user authentication scheme to fix these security flaws. As a result, their scheme fails to protect user privacy, mutually authenticate users and GWNs, and establish SN-user sessions. As part of their WSN mutual authentication and key agreement scheme, *Yang et al. (2019)* only use hash and XOR operations. According to them, their scheme protects identity, and passwords, and is resistant to attacks on stolen smart cards. The scheme is, however, susceptible to identity guessing attacks, tracking attacks, privileged insider attacks, and weak stolen smart card attacks. With the knowledge of the static value specific to some users presented in the login request message, the adversary can uncover the user's identity through offline guessing. In addition, a privileged insider may also gain access to a registered user's password by using the same method. Furthermore, we refer to weak stolen smart card attacks as exploiting the secret information contained in the stolen smart card so that the adversary can guess the user's password through offline exhaustive guessing.

Contributions:

- Analyse the existing methods that use multiple factors in authentication in wireless sensor networks (WSNs) in the literature. The advantages and disadvantages of these methods can give an overview of the limitations of the existing studies.
- To overcome the shortages in the existing method, a new authentication method is proposed in this study based on using four authentication factors: homomorphic encryption on key, timestamp, random number (nonce), and fingerprints. These factors

come with three main stages to ensure security during the authentication process between users and servers.

- The proposed method is implemented using ProVerify to verify the protocol and prove its secrecy.
- The proposed method is compared to two related studies in the literature based on many aspects to prove its effectiveness.

Article organization:

Section 'Related work' discussed the related work in detail. Section 'Preliminaries' illustrates the main preliminaries of the topic. Section 'The proposed authentication protocol' explains the proposed method. Section 'Security analysis and discussion' highlights the security analysis and challenges to the proposed method. Finally, the section 'Conclusion and future work' concludes the article and suggests some future work.

# RELATED WORK

*Chong (2022)* tackles the issue of user authentication in wireless sensor networks (WSNs) and evaluates existing two-factor authentication schemes. The authors highlight the limitations and security flaws in the schemes proposed by Das, Khan-Alghathbar, and Chen-Shih, which include vulnerabilities to various attacks and a lack of key agreement. To overcome these flaws, the authors propose a novel two-factor mutual authentication scheme with a key agreement for WSNs. They provide a security analysis and efficiency evaluation, demonstrating that their protocol is more robust and secure compared to existing schemes. The computational costs for gateway and sensor nodes are reasonable, and the proposed protocol is verified using non-monotonic cryptographic logic.

*Das et al. (2016)* proposed an efficient multi-gateway-based three-factor user authentication and key agreement scheme for hierarchical wireless sensor networks (WSNs). The scheme incorporates passwords, smart cards, and biometric information to enhance the security of user authentication. By deploying multiple gateways in the WSN, the scheme distributes the authentication workload and improves scalability. The security analysis demonstrates the scheme's resilience against various attacks, ensuring the integrity and confidentiality of user authentication. Additionally, the performance evaluation shows that the scheme has low computation and communication overhead, making it suitable for resource constrained WSN environments. The proposed scheme provides a secure and efficient solution for user authentication and key agreement in hierarchical WSNs.

In *Shin & Kwon (2019)*, address security concerns in wireless sensor networks (WSNs) for smart home systems are addressed. The authors highlight the importance of authentication and key agreement schemes to protect WSNs from security threats. They specifically focus on the scheme proposed by *Jung et al. (2017)*, which claims to offer enhanced security using biometric information. However, the authors identify several security weaknesses in *Jung et al.*'s (*2017*) scheme, including vulnerabilities related to secret key protection, session key security, user tracking attacks, information leakage, and user impersonation. Building on the work of *Jung et al. (2017)*, the authors suggest a minimal three-factor authentication and key agreement mechanism for WSNs in smart homes.

Through the use of formal security proofs and verification methodologies, they provide a thorough study of the security and effectiveness of their suggested scheme. The results demonstrate that their scheme fulfils desirable security requirements, withstands various attacks, and offers improved efficiency compared to related schemes. Overall, the article provides valuable insights into addressing security weaknesses in WSNs for smart homes and offers a secure and efficient solution for authentication and key agreement.

*Fattah et al. (2020)* gives a thorough analysis of underwater wireless sensor networks (UWSNs), focusing on the technologies' needs, most recent developments, and open research challenges. The authors emphasize the increasing interest in UWSNs for ocean surveillance, marine monitoring, and underwater target detection. They emphasize the necessity for reliable and adaptable solutions to support the quick growth of UWSNs. The article provides an analysis of the literature in the past five years, identifying key requirements and offering a taxonomy of critical elements in UWSNs, including architecture, communication, routing protocols, security, and applications. The authors also discuss the remaining challenges in UWSNs and propose future research directions. They emphasize the significance of acoustic communication in designing algorithms, protocols, and services for UWSNs. The article suggests potential areas for improvement, such as node mobility, cooperative control among underwater vehicles, high-level planning, complex network scenarios, and energy harvesting strategies. By addressing these challenges, the authors aim to enhance the performance and efficiency of UWSNs and promote their wide range of applications in underwater environments.

*Nam et al. (2015)* proposes a new two-factor user authentication scheme for wireless sensor networks (WSNs) that provides both efficiency and anonymity, it uses a lightweight sensor computation method to reduce the computational overhead on sensors. Also, a novel identity-based encryption scheme is used in purpose to achieve user anonymity. The evaluation was done through simulations, and the results show that it can provide both efficiency and anonymity in WSNs. Efficiency is achieved by offloading the computationally intensive tasks to the server. Moreover, the Anonymity property in the scheme makes it difficult for the server to track the identity of the user who is authenticated. Thus, the proposed scheme is secure against various attacks, such as replay attacks, and man-in-the-middle attacks. As this approach can offer many advantages, though it suffers from some limitations such as limited security, if an attacker can obtain the private key of the server itself, then they can impersonate any user in the network, also it requires additional communication between sensors and the server which will impact the overall network performance. In addition, it may not be scalable to many sensors, because each sensor is required to maintain a copy of the user's private key. To conclude, the proposed approach is a valuable contribution to the field of user authentication in WSNs. It provides a new way to balance the needs of security, performance, scalability, and usability.

*Jiang et al. (2015)* proposes an efficient two-factor user authentication scheme with unlink ability for wireless sensor networks (WSNs). The scheme uses two factors for user authentication: a password and a one-time password (OTP). The password is used for initial authentication, and the OTP is used for subsequent authentication. The proposed scheme achieves many of the security goals, such as identity authentication, password protection,

OTP protection, resiliency against stolen passwords, and unlink, which makes it a secure and robust authentication scheme for WSNs. Moreover, it has proven its efficiency in terms of computation and communication overhead because it uses a lightweight cryptographic algorithm and does not require the use of a trusted third party. Furthermore, the evaluation has been done through simulation, and the results show that it is secure and efficient. The proposed scheme does not rely on a specific software platform, which makes it flexible and adaptable to different WSNs. However, there is a dependency on the use of hardware in the process of generating OTP, which can hinder the performance of the WSNs that can't access such devices. Let alone that the OTP is sent in cleartext over the network, which makes the scheme vulnerable to replay attacks. Overall, the proposed scheme is a secure and efficient authentication scheme, and it's a promising approach for improving the security of WSNs.

The advent of 5G has led to the development of new technologies that enable programmable control of wireless sensor networks (WSNs). However, deploying WSNs in untrusted environments exposes them to threats of security attacks. To address this issue, *Miranda et al. (2020)* proposed a software-defined security framework. The framework combines intrusion prevention and collaborative anomaly detection systems. To facilitate lightweight intrusion prevention, IPS-based authentication is employed in the data plane. In the data plane, a collaborative anomaly detection system is integrated to provide an affordable intrusion detection solution. Moreover, a Smart Monitoring System (SMS) is utilized to reconcile accurate positive alerts caused by sensor nodes at the network edge in the control plane. The proposed model has been tested under various security scenarios and compared to other methods. The model was found to be effective in detecting and preventing intrusions, while also reducing the number of false alarms.

In *Babaeer & Al-Ahmadi (2020)* focuses on the security challenges faced by wireless sensor networks (WSNs) and proposes an efficient and secure method for data transmission and sinkhole detection. In WSNs, where sensors transmit data to a central station, security attacks, particularly sinkhole attacks, pose a threat to network integrity. Sinkhole attacks occur when a malicious node attracts other nodes by falsely advertising itself as the best path to the base station, compromising network security. To address these issues, the proposed method combines the Threshold Sensitive Energy Efficient Sensor Network (TEEN) protocol with watermarking techniques and homomorphic encryption. The use of homomorphic encryption ensures efficient and low-energy identification of sensor nodes for sinkhole detection. Concerning delay, packet delivery ratio, throughput, and average energy consumption, the suggested method performs better than earlier studies according to the results of an evaluation using the OMNET++ simulation. The lightweight watermarking and authentication techniques employed in the proposed scheme contribute to its energy efficiency. The results demonstrate that the proposed method effectively detects sinkhole attackers before they can launch an attack and ensures the integrity and authenticity of transmitted data by detecting any tampering. The article concludes by emphasizing the need for secure protocols in WSNs and highlights the effectiveness of the proposed method in providing significant security while conserving network resources. Table 1 illustrates the summary of studies in the literature.

**Table 1  Summary of studies.**

| Study Name | Description | Advantages | Limitations |
|---|---|---|---|
| (*Chong, 2022*) Two-factor mutual authentication with key agreement in wireless sensor networks | Evaluates existing two-factor authentication schemes in WSNs and propose a novel scheme with key agreement. | Addresses limitations and security flaws in existing schemes Provides robustness and security Reasonable computational costs. Verified using non-monotonic cryptographic logic. | May require further evaluation and testing in different scenarios or network configurations. |
| (*Das et al., 2016*) An efficient multi-gateway-based three-factor user authentication and key agreement scheme in WSNs | Proposes a three-factor authentication scheme using passwords, smart cards, and biometrics for hierarchical WSNs. | Enhances security through multiple factors Distributes authentication workload with multiple gateways Low computation and communication overheads. | N/A |
| (*Shin & Kwon, 2019*) A lightweight three-factor authentication and key agreement scheme in WSNs for smart homes | Addresses security weaknesses in authentication schemes for WSNs in smart homes and proposes an improved scheme | Identifies and addresses security weaknesses Offers improved security and efficiency Provides detailed security analysis and proofs | Specific to WSNs in smart homes - Limited comparison with other related schemes Absence of implementing and conducting experiments on real devices to assess the performance of the suggested scheme |
| (*Fattah et al., 2020*) A survey on underwater wireless sensor networks: requirements, taxonomy, recent advances, and open research challenges | Presents a comprehensive survey on UWSNs, including requirements, recent advancements, and open challenges | Provides an overview of UWSN requirements and advancements Offers a taxonomy of critical elements in UWSNs Identifies open research challenges | Limited to survey and analysis May require further research to address identified challenges Limited timeframe (past five years) |
| (*Nam et al., 2015*) Efficient and anonymous two-factor user authentication in wireless sensor networks: achieving user anonymity with lightweight sensor computation | Proposes a new two-factor user authentication scheme for wireless sensor networks (WSNs) that provides both efficiency and anonymity | Efficiency: The lightweight sensor computation method reduces the computational overhead on sensors. Anonymity: the server cannot track the identity of the user who is authenticated. Security: secure against various attacks. Scalability: scalable to a large number of users and sensors. | Limited security, the massive communication between sensors and the server overhead the overall network, and not be scalable to many sensors, |
| (*Jiang et al., 2015*) An efficient two-factor user authentication scheme with unlink ability for wireless sensor networks | Proposes an efficient two-factor user authentication scheme with unlink ability for wireless sensor networks (WSNs). | Achieves many of the security goals which makes it a secure and robust authentication scheme for WSNs. Efficient in terms of computation and communication overhead. Does not rely on a specific software platform, which makes it adaptable to different WSNs. | Dependency on the use of hardware in generating OTP, which hinders the performance of the WSNs that can't access such devices. OTP is sent in cleartext making the scheme vulnerable to replay attacks. |

**Table 1** (*continued*)

| Study Name | Description | Advantages | Limitations |
|---|---|---|---|
| (*Miranda et al., 2020*) A collaborative security framework for software-defined wireless sensor networks | Proposal of a software-defined security framework for WSNs | Combines collaborative anomaly detection and intrusion prevention systems. Data-plane intrusion prevention system that is lightweight. A low-cost intrusion detection system close to the data plane. | It is a theoretical study, which means that it is not yet clear how well the framework would work in practice. It does not consider all possible attack scenarios. |
| (*Babaeer & Al-Ahmadi, 2020*) Efficient and secure data transmission and sinkhole detection in a multi-clustering wireless sensor network based on homomorphic encryption and watermarking | Proposes a lightweight, secure method for data transmission and sinkhole detection in a multi-clustering wireless sensor network. | Uses homomorphic encryption to ensure data integrity and confidentiality. Uses watermarking to detect sinkhole attacks. Uses a multi-clustering approach to improve performance. | Not yet tested in a real-world environment. May not be effective against all types of sinkhole attacks. |

## PRELIMINARIES

This section will illustrate the most common terminologies that are used in the proposed method.

### Homomorphic encryption

Homomorphic encryption is a type of encryption that allows computations to be performed on encrypted data without first decrypting it (*Acar et al., 2018*). The resulting computations are left in an encrypted form which, is identical to the outputs produced by computations on unencrypted data when they are decrypted. Homomorphic encryption is a promising technology with a wide range of potential applications. However, it is still under development, and some challenges need to be addressed before it can be widely adopted, such as Computational complexity and expense. There are many applications of Homomorphic encryption, for example:

### *Secure data sharing*

It allows to share the sensitive data without faring from being compromised, for example, a doctor could share a patient's medical records with another doctor without having to decrypt the records (*Singh & Saxena, 2022*).

### *Anonymous cloud computing*

Users can encrypt their data and upload it to the cloud, while the processing takes place without a need to decrypt the data (*Mo et al., 2019*).

### *Improve financial services*

Banks can use homomorphic encryption to encrypt the transactions and then process them without knowing the unencrypted data (*Ali, Ally & Dwivedi, 2020*).

### Timestamp

A timestamp is a data element utilized in cryptographic protocols to record the time of a specific event, serving various purposes such as ensuring timeliness and preventing replay attacks. In mainstream cryptographic standards, timestamps are commonly employed to

enhance the security of protocols (*Tewari & Gupta, 2020*). Incorporating timestamps into cryptographic protocols offers several security benefits:

### Prevention of replay attacks

Replay attacks involve capturing a valid message and replaying it later. Timestamps effectively mitigate such attacks by restricting the acceptance of messages within a specific time window (*Parameswarath & Sikdar, 2022*).

### Assurance of timeliness

Timestamps provide assurance that data was created or modified at a precise moment in time. This feature proves valuable in applications like electronic voting and digital contracts. Timestamps play a crucial role in enhancing the security of cryptographic protocols by guaranteeing timeliness and safeguarding against unauthorized access through replay attacks (*Zhang et al., 2023b*).

## Generate random numbers

Random numbers are unpredictable and not dependent on previous numbers or data. They are produced by random number generators (RNG), which are devices or algorithms that create a series of statistically independent and uniformly distributed numbers. RNGs are utilized in a variety of applications, including cryptography, gaming, and simulation (*Sangeetha et al., 2023*).

## Biometrics

Biometrics is the measurement and examination of physical traits of the human body to uniquely identify a person (*Ismatillayev, 2022*). The data is collected and stored by biometric systems using a variety of techniques, including voiceprints, facial recognition, and fingerprints. Then, this information can be utilized to authenticate users or to keep tabs on their whereabouts. Some biometrics examples include the following:

- One of the most popular biometrics used for identification is the fingerprint. They can be used to log into accounts, unlock gadgets, and gain access to secure locations because they are personal to each person (*Alhmiedat, 2023*).
- Facial recognition: Another widely used biometric for identification and access control is facial recognition. To confirm someone's identity, it compares their face to a saved photograph (*Rodrigues et al., 2021*).
- Voiceprints: Each person has a distinct voiceprint, which can be used to identify and authenticate users. They are frequently employed in call centres and other settings where it is crucial to confirm the caller's identity (*Boulmaiz et al., 2020*).
- Iris recognition: A biometric that uses a person's individual iris patterns to identify them. It is a biometric that is exceptionally accurate and is frequently utilized in high-security applications (*Lei et al., 2022*).
- Venous recognition: A biometric that uses a person's own venous pattern to identify them. Despite being a relatively recent biometric, because of its convenience and precision, it is gaining popularity (*Prabhu, Muthu Kumar & Ahilan, 2023*).

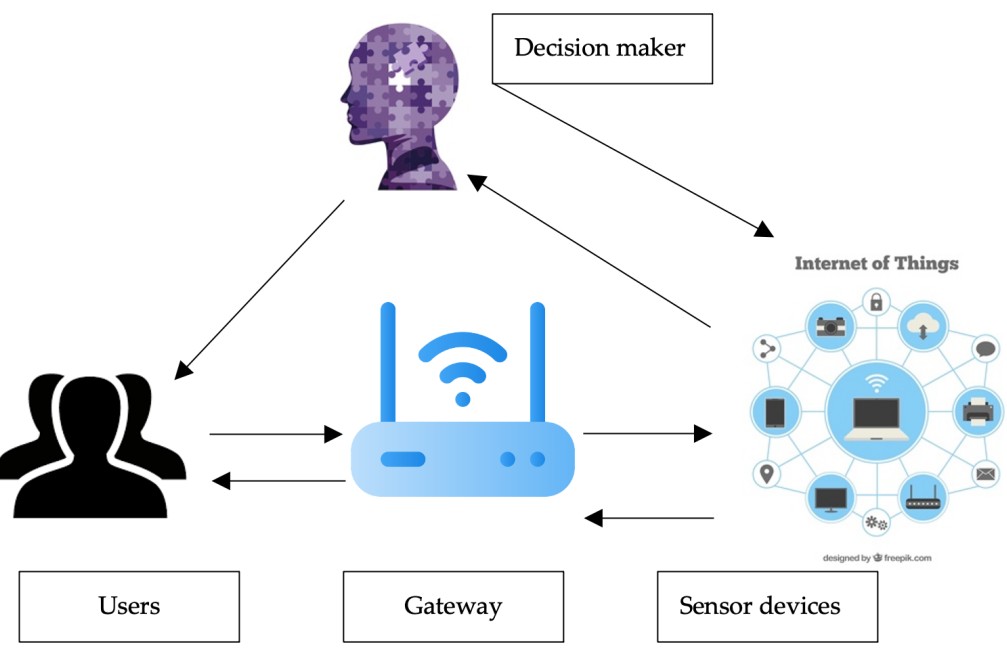

**Figure 1** **The proposed method model.** Figure source credits: Head and shoulders image source credit: simpleicon.com. Epilepsy day profile image: Freepik. Network background with devices image: Freepik. Internet of things icons created by apien - Flaticon.

## THE PROPOSED AUTHENTICATION PROTOCOL

The proposed method is composed of four entities as it is shown in Fig. 1:

- User: The user applies to the gateway to use the sensing devices. Users should be registered and authenticated to use the IoT services and sensors.
- Gateway: All users and sensing devices need to be registered with the gateway. The role of the gateway is to organize the process and ensure authentication is applied and regulated between both users and sensing devices. The gateway keeps all the information about the registered users to communicate with the sensors.
- Sensors: Sensor devices can be deployed in various IoT environments such as smartwatches and smart vehicles. Data is collected and provided to the users and this data can be used to execute commands. The main limitation of most sensing devices is constrained power.
- Decision maker: Depending on the results of the readings from the sensors, the decision maker uses an artificial intelligence model to decide the decision of the results. Federated learning is suggested to be used in this stage.

In this section, we propose a new four-factor authentication user protocol.-factor authentication user protocol. As the number of authentication factors increases, the security level becomes better. The four factors that are used are homomorphic encryption on key, timestamp, random number (nonce), and personal biometrics such as a fingerprint.

**Table 2 Illustrates the notation used in the proposed method.**

| Symbol | Description |
| --- | --- |
| $Reg_r$ | Register request |
| $U_i$ | Username |
| TS | Timestamp |
| N | Random number |
| $ID_i$ | Identity number |
| $P_w$ | Password |
| H | Hash function |
| $E_n$ | encryption |
| $r_s$ | Gateway |
| DB | Database |
| $F_{ID}$ | User Fingerprint |
| $H_e$ | Homomorphic encryption |
| $Ul_s$ | Ultrasonic sensor |
| Pu | Public key |
| $P_v$ | Private key |

Table 2 illustrates the notations that are used in the proposed method. There are three main stages in the proposed method:

**Phase1: Registration**

The registration phase starts when the user registers for the first time with the remote server using identity and passwords. The registration phase enables users to use IoT devices connected with sensor devices to perform various tasks. Registration is done with the gateway. Table 1 illustrates the notation used in this study.

  Initially, the user put the identity number and encrypted password as follows:

1. The user will start to insert their own (IDi).

  Encrypt both the (IDi ), password, and timestamp (TS) using the hash function, so the credentials will be sent to the gateway in encrypted format as follows: h(IDi || Pw||TS) (rs ) checks the database that is saved on it in an encrypted format. If the (IDi) is already registered, then login is completed and the user is authenticated.

  if the (IDi) is new, then the registration phase is completed

  The usage of the hash function can overcome the security flaw when sending the credentials without the hash function.

  Inputs: IDi h(IDi || Pw) : rs

  rs verifies the request

  rs checks DB: if IDi in DB and the credentials are matched, then log in —à authentication success

  else IDi registration.

  Figure 2 shows the registration process between both the user and sensor devices with the gateway.

  The following are the steps of a handshake between the user and the gateway:

2. The user sends the request to start negotiation to the gateway such as a ping.

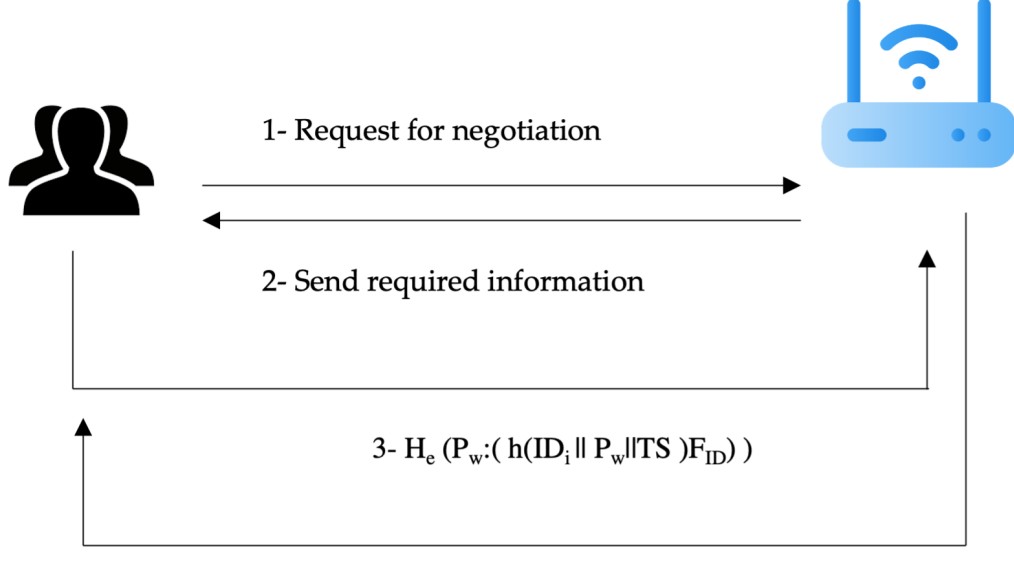

**Figure 2 Registration process negotiations.** Figure source credits: Head and shoulders image source credit: simpleicon.com. Network background with devices image: Freepik.

The gateway way will check the registered ID of the user and if the user is a new user, then the gateway will send a request to the user to provide the required information to complete the registration process.

The user will send the required information in a secure form to preserve the integrity and confidentiality of the information. The registration process should use many factors such as ID, password, timestamp, and features of fingerprint biometrics. The transfer of this information needs to be secure. Thus, both homomorphic encryption and hash are used to ensure the privacy and confidentiality of this information. The computation form is sent to the gateway.

The gateway will receive the required information, register it, and save it to the database in an encrypted format. The gateway will send the completion of registration to the user.

The biometrics will be captured by the special reader as a sensor. Not the whole block of the fingerprint will be saved at the gateway, only the features of the biometrics to decrease the overhead on the gateway database.

## Phase2: Authentication (login)

The popular RSA encryption system is an illustration of a partial homomorphic encryption method. One of RSA's properties is that can get the encryption of $a \cdot b$ by multiplying the coding of $a$ and the coding of $b$. Since data is encrypted on clouds, homomorphic encryption finds its greatest use in the cloud computing industry. In addition to the many completely homomorphic cryptosystems that are accessible today, there are a number of partially homomorphic cryptosystems as well. Any cryptosystem can accidentally be manipulated

and is vulnerable to several kinds of assaults. A cautious approach to this issue may be made using homomorphic operations, which carry out calculations safely. In the proposed method, RSA is used as a partially homomorphic cryptosystem. Partially homomorphic encryption is limited to a particular kind of single algebraic operation (multiplication, addition, *etc.*); examples of this include the RSA method. Partially homomorphic encryption has the benefit of being relatively simple, which allows for significant processing reduction in most situations. On the other hand, it has the drawback of being limited to a single type of operation.

If two messages RSA encryption is used to protect m1 and m2, and the encrypted key is K1, where n = p ∗ q, p, and q are two huge prime integers. When these encrypted messages are added together, the homomorphic characteristic is demonstrated as follows:

$$Encrypt(m1) * Encrypt(m2) = (m1^d(modn)) * (m2^d(modn))$$
$$= (m1 * m2)d(modn) = Encrypt(m1 * m2). \tag{1}$$

Suppose that the user already has registered and has credentials: username $U_i$ and password $P_w$. The user now login to the system by sending the credentials encrypted using homomorphic encryption, and timestamp. The two keys that are used in the fully homomorphic encryption are generated and saved in the gateway. The encryption of the credentials will be as follows:

$$H_e(P_u(ID_i \| P_w \| TS \| Ul_s \| P_v)). \tag{2}$$

The encrypted credentials are sent to the server in order to do the mutual authentication and check the user identity. When the server receives the encrypted credentials, then it needs to decrypt it first. The decryption process uses the server's private key to decrypt the data. As partial homomorphic encryption is used in the proposed method, thus the RSA algorithm is considered in both encryption and decryption. After the credentials are decrypted, the server will consider the timestamp check the credentials that have been saved in the service and make the decision if the user credentials are authenticated and authorised or not. The decryption of the credentials will be as follows:

$$D_h(H_e \| TS). \tag{3}$$

## Phase3: Password and biometrics verification

Every human has a different fingerprint even twins. After the login process is completed, then to improve the security level another verification should be done. The ultrasonic fingerprint sensor is used to extract the features of the fingerprint. There are four different types of fingerprint sensors: optical scanners, capacitance scanners, ultrasonic scanners, and thermal scanners. The reason for using the ultrasonic fingerprint feature is it is more secure regards the usage of 3D capture. Figure 3 shows the format of the ultrasonic sensor. The ultrasonic sensor is a type of sensor that converts physical quantities into electrical format and vice versa. The main concept of this type of sensor is using a sound wave's

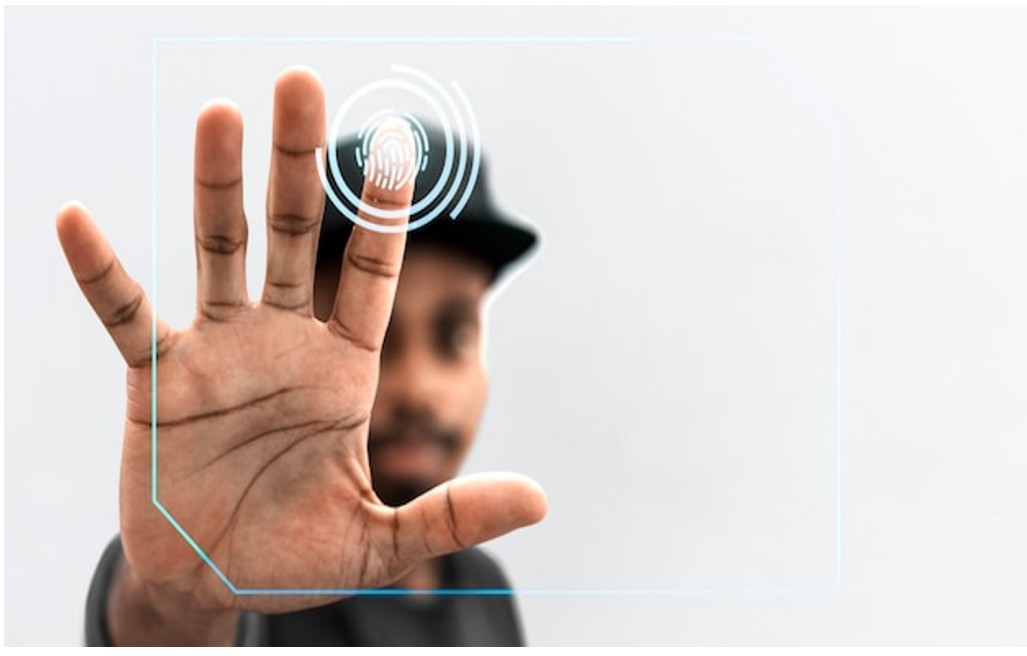

**Figure 3** **Ultrasonic fingerprint sensor.** Figure source credit: Identification scanning system image: Freepik.

reflection to capture the fingerprint points with a specific frequency. While this type uses ultrasonic sound of 20,000 hertz it is called ultrasonic sensor.

$$H(F_{ID}\|TS). \tag{4}$$

In the proposed method the usage of fingerprints is to increase the security level. Biometrics have many features that cannot be forgotten or lost and have a low chance of being forged or shared. An ultrasonic sensor is a powerful tool in many applications such as in the medical dialogistic sector. it uses 3D ultrasound which is more efficient information on human anatomy than 2D ultrasound. The concept of this type of sensor stems from transmitting an ultrasonic pulse of inaudible sound wanes related to the finger that presses the scanner and reads. The information is passed to the receiver where the reading is mapping out the ridges on the fingerprint to measure the three-dimensional rendering.

To improve computer security, biometric data can be fed into common cryptography methods with the help of fuzzy extractors. In this sense, "fuzzy" refers to the ability to obtain the fixed values needed for cryptography from values that are similar to the initial keys but not exact, all while maintaining the necessary level of security. One use is to employ the user's biometric inputs as a key for protecting and authenticating user records. In the proposed method, the methodology for biometric verification is based on a fuzzy extractor. Unlike current fuzzy extractor approaches that need linear-time calculation, the suggested protocol may identify an individual with a fixed computational cost.

Federated learning (FL) is used in the decision-making process to decide if the user is authenticated or not. FL is a distributed secure machine learning model where a central

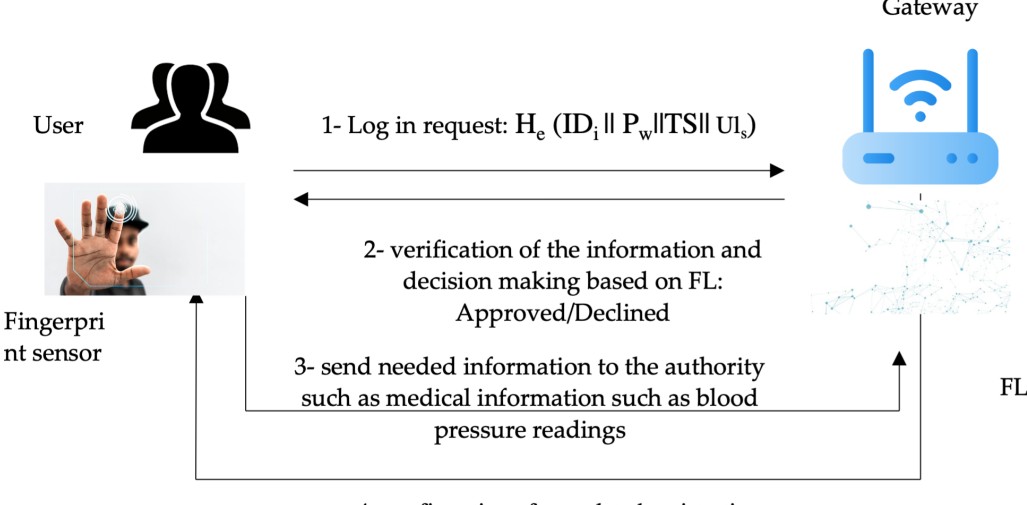

**Figure 4   Authentication process in the proposed method.** Figure source credits: Head and shoulders image source credit: simpleicon.com. Network background with devices image: Freepik. Identification scanning system image: Freepik. Technology vector image: Freepik.

server can communicate with many devices such as smartphones. It enables limited storage and computation processes. Figure 4 illustrates the authentication process in the proposed method.

## SECURITY ANALYSIS AND DISCUSSION

The proposed authentication protocol leverages a four-factor authentication mechanism that includes homomorphic encryption on key, timestamp, random number (nonce), and personal biometrics such as a fingerprint. The protocol spans three main stages, registration, authentication, and biometrics verification, across four entities, User, Gateway, Sensors, and Decision Maker. In this section, we used ProVerify (*Blanchet, 2016*) to first prove protocol authentication and then protocol secrecy. ProVerif is a widely acclaimed tool for the verification of cryptographic protocols within the formal model. It is an automatic cryptographic protocol verifier in the Dolev-Yao model, developed by Bruno Blanchet. The tool allows for the modelling and analysis of protocols in an abstract way, capturing their essential characteristics without delving into the low-level details of cryptographic algorithms. ProVerif's automatic capabilities are leveraged to verify a wide range of security properties, including confidentiality, integrity, and authentication. It supports various cryptographic primitives and can handle an unbounded number of protocol sessions, providing significant advantages in analysing real-world cryptographic protocols. The underlying principle of ProVerif relies on translating the protocol into a set of Horn clauses and resolving their satisfiability. ProVerif's efficiency and effectiveness have led to its wide

**Figure 5** Authentication result.

```
--  Process 1 (that is, process 0, with let moved downwards):
{1}new PKGW: pu;
{2}new Pwi: PW;
{3}new IDi: ID;
{4}new PKGW_1: pu;
{5}new PRGW: pv;
(
    {6}!
    {10}new ts: TS;
    {9}let Pwi_1: PW = Pwi in
    {7}let IDi_1: ID = IDi in
    {11}let rq: request = hash(IDi_1,Pwi_1,ts) in
    {12}out(c, rq);
    {13}in(c, x: pu);
    {8}let PKGW_2: pu = PKGW_1 in
    {14}if (x = PKGW_2) then
    {15}event acceptsGate(PKGW_2);
    {16}new cr_1: credentials;
    {17}let c': credentials = He(cr_1,PKGW_2) in
    {18}out(c, c');
    {19}event termUser(x)
) | (
    {20}!
    {25}in(c, reqq: request);
    {26}new ts2: TS;
    {22}let Pwi_2: PW = Pwi in
    {21}let IDi_2: ID = IDi in
    {27}let reqq2: request = hash(IDi_2,Pwi_2,ts2) in
    {28}if (reqq = reqq2) then
    {29}event acceptsUser(reqq2);
    {23}let PKGW_3: pu = PKGW_1 in
    {30}out(c, PKGW_3);
    {31}in(c, x_1: credentials);
    {24}let PRGW_1: pv = PRGW in
    {32}let yy: credentials = Hd(x_1,PRGW_1) in
    {33}new xx: credentials;
    {34}if (yy = xx) then
    {35}event termGate(yy)
)

-- Query not attacker_credentials(cr[]) in process 1.
Translating the process into Horn clauses...
Completing...
Starting query not attacker_credentials(cr[])
RESULT not attacker_credentials(cr[]) is true.

--------------------------------------------------------
Verification summary:

Query not attacker_credentials(cr[]) is true.

--------------------------------------------------------
```

**Figure 6  Secrecy result.**

adoption in cryptographic research, making it a standard tool for protocol analysis (*Rashid & Rehmani, 2016*). Figure 5 shows the authentication result.

Based on the result, the protocol achieves strong authentication because the protocol successfully verifies the identity of the user. This means that the combination of the four factors (homomorphic encryption, timestamp, nonce, and fingerprint) effectively ensures that only authorized users can access the system. Also, the protocol is resistant to impersonation: ProVerif analysed the protocol against potential impersonation attacks, and a successful verification shows that such attacks are mitigated. Moreover, the protocol is sound (Implementation Validity): The correctness of the complex interactions and

cryptographic processes in the protocol would be validated by ProVerif, indicating that the design is logically sound. Figure 6 shows the protocol secrecy results.

The protocol's achievement of confidentiality is substantiated by the comprehensive utilization of homomorphic encryption, hashing, and other cryptographic measures, all of which have been thoroughly validated by ProVerif. This ensures that data remains inaccessible to unauthorized entities, preserving its confidentiality. Additionally, the protocol demonstrates resilience against eavesdropping: its design effectively safeguards against potential eavesdroppers, preventing them from extracting any useful information even if they manage to intercept the data. Moreover, the protocol's approach to Biometric Data Security has been meticulously crafted to enhance the safety of sensitive information. Particularly, the handling and secure storage of delicate biometric data, such as fingerprints, have been verified to minimize the risk of exposure or misuse, further fortifying the overall security framework.

Additionally, we selected two closely related and promising schemes for a comparative analysis with our protocol. Table 3 presents the results of this comparison, offering a detailed and insightful evaluation of how our protocol stands in relation to these selected schemes. Table 3 illustrates the comparison of the proposed method with the other two related works in the literature.

Based on the results, "Our Protocol" stands out with its advanced four-factor authentication mechanism, which includes homomorphic encryption, timestamp, nonce, and fingerprint biometrics. This comprehensive approach provides a multi-layered security barrier, making it significantly more robust than the three-factor systems used in the other schemes. In terms of security stages, "Our Protocol" covers registration, authentication, and biometrics verification, ensuring thorough security checks at each stage.

In contrast, "Scheme (*Das et al., 2016*)" employs a three-factor authentication scheme, involving passwords, smart cards, and biometrics, and goes through several phases, including pre-deployment, user registration, login, authentication and key agreement, and password and biometric updates. "Scheme (*Shin & Kwon, 2019*)" similarly utilizes a combination of smart card information, user password, biometric information, user identity, and a unique user identifier. Its security stages encompass system setup, user registration, login, authentication, and password change.

The entities involved in "Our Protocol" include users, gateways, sensors, and decision-makers, demonstrating a comprehensive engagement of various components in the network. This is slightly more expensive compared to "Scheme (*Das et al., 2016*)," which involves user gateways, gateway nodes, and smart cards, and "Scheme (*Shin & Kwon, 2019*)," which includes users, home gateway nodes, and sensor nodes.

Verification tools are crucial in assessing the reliability of these protocols. "Our Protocol" is verified using ProVerif, known for its effectiveness in cryptographic protocol verification. On the other hand, "Scheme (*Das et al., 2016*)" uses AVISPA for its verification, while "Scheme (*Shin & Kwon, 2019*)" is evaluated using both BAN logic and AVISPA.

In terms of authentication strength, "Our Protocol" is rated with strong authentication, an indicator of its robust security measures. This is paralleled by "Scheme (*Shin & Kwon, 2019*)," but contrasts with "Scheme (*Das et al., 2016*)," which is rated as having

Alsumayt et al. (2024), *PeerJ Comput. Sci.*, DOI 10.7717/peerj-cs.2091

**Table 3 Comparison of the proposed method with other studies.**

| scheme name | Authentication factors | Security stages | Entities involved | Verification tool used | Authentication strength | Impersonation resistance | Protocol soundness | Confidentiality measures | Eavesdropping resilience |
|---|---|---|---|---|---|---|---|---|---|
| Our protocol | Homomorphic encryption, timestamp, nonce, fingerprint | Registration, Authentication, Biometrics Verification | User, Gateway, Sensors, Decision Maker | ProVerif | Strong | High | Sound | Homomorphic encryption, hashing | High |
| Scheme (*Das et al., 2016*) | three-factor authentication scheme using passwords, smart cards, and biometrics | Pre-deployment stage User registration phase Login phase Authentication and key agreement phase Password and biometric update phase | User gateway Gateway nodes Smart cards | AVISPA | Medium | Medium | Sound | Hashing | Low |
| Scheme (*Shin & Kwon, 2019*) | Smart card information (SCi), User password (PWi), Biometric information (Bioi), User identity (IDi), A unique user identifier (ui) | System Setup, User Registration, Login, Authentication, Password Change | User (Ui), Home Gateway Node (HG), Sensor Nodes (Sj) | BAN AVISPA | Strong | LOW | Sound | ECC (Elliptic Curve Cryptography), Hashing | High |

medium authentication strength. Additionally, "Our Protocol" exhibits high resistance to impersonation attacks, surpassing the medium resistance of "Scheme (*Das et al., 2016*)" and the low resistance of "Scheme (*Shin & Kwon, 2019*)."

All three protocols are considered sound, indicating their logical consistency and effectiveness. However, the confidentiality measures of "Our Protocol," which include homomorphic encryption and hashing, provide a higher security level compared to the simple hashing in "Scheme (*Das et al., 2016*)" and the combination of ECC and hashing in "Scheme (*Shin & Kwon, 2019*)." Furthermore, "Our Protocol" and "Scheme (*Shin & Kwon, 2019*)" both show high resilience against eavesdropping, while "Scheme (*Das et al., 2016*)" falls behind with low eavesdropping resilience.

Overall, "Our Protocol" demonstrates superiority in several key areas. Its robust multi-factor authentication, combined with strong verification tools and comprehensive security stages, provides a well-rounded and secure solution. The high impersonation resistance and strong authentication strength make it a preferable choice for secure network environments. The inclusion of advanced cryptographic measures, such as homomorphic encryption and the high resilience against eavesdropping, further reinforce its position as a leading protocol in the realm of network security.

## CONCLUSION AND FUTURE WORK

The IoT helps to create a feasible Internet infrastructure and multiple sensor devices. Due to the intensive use of the Internet, there is a serious concern regards the security aspect. Many factors need to be considered in the system to be secure starting from the CIA triad: Confidentiality, Integrity, and Availability, to reach authentication, authorisation, and non-repudiation. According to the idea of ubiquitous computing, services may be accessible at any time and from any location by using IoT-enabled machines. Remote users can quickly log in to enjoy the services that are offered and profit from them. One of the early uses of WSNs is authentication. WSNs are used for a variety of tasks, including combat surveillance, healthcare delivery, diagnostic evaluation, and many more. To address the weaknesses and safety issues with user authentication systems, several security designs have been proposed in the literature. In this article, we proposed a novel method to complete authentication in wireless sensor networks based on four factors. Homomorphic encryption on key, timestamp, random number (nonce), and fingerprint are used in this method. The implementation results of the proposed method confirm the effectiveness of the method to authenticate users in systems. The proposed method can be usable in many sectors and organisations such as banking, healthcare applications, educational purposes, purchasing online, and so on. We assumed that based on the readings, decision-makers based on federated learning will be done to decide on whether the authentication is successful or not. For future work, other biometrics will be used and the comparison between the performance of each biometric method will be done to evaluate the better method. In addition, other encryption methods for keys can be used instead of homomorphic encryption to choose the best method with a strong performance rate.

### Funding

The SAUDI ARAMCO Cybersecurity Chair funded this article. The funders had no role in study design, data collection and analysis, decision to publish, or preparation of the manuscript.

### Grant Disclosures

The following grant information was disclosed by the authors:
The SAUDI ARAMCO Cybersecurity Chair.

### Competing Interests

The authors declare there are no competing interests.

### Author Contributions

- Albandari Alsumayt conceived and designed the experiments, performed the experiments, analyzed the data, performed the computation work, prepared figures and/or tables, authored or reviewed drafts of the article, and approved the final draft.
- Majid Alshammari conceived and designed the experiments, performed the experiments, performed the computation work, authored or reviewed drafts of the article, and approved the final draft.
- Zeyad M. Alfawaer conceived and designed the experiments, prepared figures and/or tables, authored or reviewed drafts of the article, and approved the final draft.
- Fahd N. Al-Wesabi conceived and designed the experiments, authored or reviewed drafts of the article, and approved the final draft.
- Nahla El-Haggar conceived and designed the experiments, analyzed the data, authored or reviewed drafts of the article, and approved the final draft.
- Sumayh S. Aljameel conceived and designed the experiments, analyzed the data, authored or reviewed drafts of the article, and approved the final draft.
- Sarah Albassam analyzed the data, prepared figures and/or tables, and approved the final draft.
- Shahad AlGhareeb analyzed the data, prepared figures and/or tables, and approved the final draft.
- Nouf Mohammed Alghamdi analyzed the data, performed the computation work, prepared figures and/or tables, and approved the final draft.
- Nawir Aldossary performed the computation work, prepared figures and/or tables, and approved the final draft.

### Data Availability

The code is available in the Supplemental Files.

### Supplemental Information

Supplemental information for this article can be found online at http://dx.doi.org/10.7717/peerj-cs.2091#supplemental-information.

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
