# Peer review of "Efficient security level in wireless sensor networks (WSNs) using four-factors authentication over the Internet of Things (IoT)"

_PeerJ Computer Science, doi:10.7717/peerj-cs.2091_

## Round 0.1 · original submission · Major Revisions

The comments of the reviewers who evaluated your manuscript. We ask that you make major changes to your manuscript based on those comments, before uploading the final files.

**Language Note:** The review process has identified that the English language must be improved. PeerJ can provide language editing services - please contact us at [email protected] for pricing (be sure to provide your manuscript number and title). Alternatively, you should make your own arrangements to improve the language quality and provide details in your response letter. – PeerJ Staff

Reviewer 1 ·

Basic reporting

No comment

Experimental design

No comment

Validity of the findings

No comment

Additional comments

1. The performance evaluation section is an integral part of every demonstrated authentication protocol which shows the comparative studies. This seems to be grossly missing in the scheme.
2. The paper is hard to understand as the mutual authentication phase is not pictorially depicted.
3. On line 352, the timestamp seems to be mistakenly shown as Pw, i.e. “the timestamp (Pw )”, however it must be password.
4. The authentication procedures sections need further elaboration.
5. The finger biometric FID is employed without fuzzy extractor which may render the scheme inapplicable due to de-synchronization.
6. Why timestamp is considered as the fourth factor? Did you count smart card as a factor in this scheme?
7. This paper shows serious concerns over the quality, as this does not present formal security analysis as well as performance analysis sections. Besides, the methodology is not properly demonstrated. I think the quality of this paper does not qualify for this journal as there are many serious concerns in the scheme.

Reviewer 2 ·

Basic reporting

no comment

Experimental design

no comment

Validity of the findings

no comment

Additional comments

no comment

Reviewer 3 ·

Basic reporting

Review Comments
Journal: PeerJ Computer Science
Date of Review: Dec 11, 2023

Decision: Major Revision
Efficient security level in wireless sensor networks (WSNs) using four-factors authentication over the Internet of Things (IoT)

This technical article, “Efficient security level in wireless sensor networks (WSNs) using four-factors authentication over the Internet of Things (IoT) ,” is a study that discusses the increasing use of technology, particularly IoT and Wireless Sensor Networks, to enhance daily tasks and business operations, especially in the context of COVID-19. The focus is on addressing security concerns in these networks by proposing a robust four-factor authentication protocol, with simulation results demonstrating its effectiveness in securing communications.

Here are some of the concerns I observed while reviewing your article:

Issue #1, many users get annoyed by the two factor authentication and it is too much to catty the cellphone with you always. Do you think adding the factors would lead to better security.
Issue #2, there are should be the four phases could you please elaborate more on the 4th one
Phase1: Registration
Phase2: Authentication (Log in)
Phase3: Password and biometrics verification
Issue #3, the study lacks empirical data or experiments please back up your study with data and experimental results to prove how strong your hypothesis is?
Issue #4, The paper requires significant improvements before publication.
Issue #5, The paper will be accepted after addressing the suggestive changes.

Experimental design

Issue #2, there are should be the four phases could you please elaborate more on the 4th one
Phase1: Registration
Phase2: Authentication (Log in)
Phase3: Password and biometrics verification

Validity of the findings

Issue #3, the study lacks empirical data or experiments please back up your study with data and experimental results to prove how strong your hypothesis is?

Additional comments

This paper requires severe improvement.

·

Basic reporting

The paper titled "Efficient security level in wireless sensor networks (WSNs) using four- factors authentication over the Internet of Things (IoT) is a novel writing. This study proposed a new protocol based on using four factors of authentication to improve the security level in WSN to secure communications. The writing style requires some improvement in sentence formation for better understanding the manuscript.

Experimental design

The objective and motivation for the research has been very well stated in the introduction part. But needs clarification on the following:
1. These objective need more clarification on the novelty of proposed method.
2. The proposed methodology seems very basic and need more detailed specifications.
3. Novelty of research work and finding needs more discussion in result and discussion section.

Validity of the findings

The authors adequately evaluated their work, but all simulation results needs more clear justification and should be supported by some comparative literature work.

Additional comments

My suggestion is acceptance with major revisions

---

## Round 0.2 · Minor Revisions

All concerns raised by the reviewers have been partially addressed. The manuscript still needs further clarification regarding a formal security analysis of the proposed methodology, performance comparisons with previously reported approaches, and specifying how RSA and homomorphic encryption have been employed in the proposed protocol. These issues require a minor revision. If you are prepared to undertake the work required, I would be pleased to reconsider my decision. Please submit a list of changes or a rebuttal against each concern when you submit your revised manuscript.

Reviewer 1 ·

Basic reporting

No comment

Experimental design

No comment

Validity of the findings

No comment

Additional comments

The authors have not revised the scheme accordingly. There is no formal analysis in the scheme. The description of proposed scheme lacks pictorial representation. There are no performance comparison tables. I do not see any novelty from security view point. The author did not even specify how RSA and homomorphic encryption has been employed in the proposed protocol.

Reviewer 3 ·

Basic reporting

This paper is good to go; the suggested changes have been incorporated.

Experimental design

This paper is good to go; the suggested changes have been incorporated.

Validity of the findings

This paper is good to go; the suggested changes have been incorporated.

·

Basic reporting

The paper titled "Efficient security level in wireless sensor networks (WSNs) using four- factors authentication over the Internet of Things (IoT) is a novel writing. This study proposed a new protocol based on using four factors of authentication to improve the security level in WSN to secure communications .Authors have dome all the major amendments in manuscript as suggested in first review. So as per my opinion paper is suitable for publication in its present state.

Experimental design

The objective and motivation for the research has been very well stated in the introduction part.
As per suggestions:
1.Objectives/Contributions has updated.
2.Mehodology is also updated.
As per suggestions , Authors have extensively revised and improved the "Security Analysis" section, which is now titled "Security Analysis and Discussion." In this updated section, They have included a thorough discussion emphasizing the novelty and significance of our research findings.
Additionally, They have integrated a new comparison table within this section. This table provides a direct comparison between our protocol and other comparative studies, highlighting the unique features and improvements in work introduces.

Validity of the findings

All suggestions incorporated properly.

Additional comments

No

---

## Round 0.3 · accepted · Accept

I am pleased to inform you that your work has now been accepted for publication in PeerJ Computer Science.

Please be advised that you are not permitted to add or remove authors or references post-acceptance, regardless of the reviewers' request(s).

Thank you for submitting your work to this journal. On behalf of the Editors of PeerJ Computer Science, we look forward to your continued contributions to the Journal.

With kind regards,